# Spatial tuning of face part representations within face-selective areas revealed by high-field fMRI

**Jiedong Zhang[1,2]\*, Yong Jiang[1,2], Yunjie Song[1,2], Peng Zhang[1,2], Sheng He[1,2,3]\***

[1]Institute of Biophysics, Chinese Academy of Sciences, Beijing, China; [2]University of Chinese Academy of Sciences, Beijing, China; [3]Department of Psychology, University of Minnesota, Minneapolis, United States

**Abstract** Regions sensitive to specific object categories as well as organized spatial patterns sensitive to different features have been found across the whole ventral temporal cortex (VTC). However, it is unclear that within each object category region, how specific feature representations are organized to support object identification. Would object features, such as object parts, be represented in fine-scale spatial tuning within object category-specific regions? Here, we used high-field 7T fMRI to examine the spatial tuning to different face parts within each face-selective region. Our results show consistent spatial tuning of face parts across individuals that within right posterior fusiform face area (pFFA) and right occipital face area (OFA), the posterior portion of each region was biased to eyes, while the anterior portion was biased to mouth and chin stimuli. Our results demonstrate that within the occipital and fusiform face processing regions, there exist systematic spatial tuning to different face parts that support further computation combining them.

## Editor's evaluation

How the brain is organized to represent various concepts has long been a central cognitive neuroscience research topic. Zhang and colleagues investigated the spatial distribution of feature tuning for different face-parts within face-selective regions of human visual cortex using ultra-high resolution 7.0 T fMRI. The findings complement non-human primate studies of face-selective patches and will be of interest to psychologists and system neuroscientists.

\*For correspondence:
zhangjiedong@gmail.com (JZ);
hes@ibp.ac.cn (SH)

**Competing interest:** The authors declare that no competing interests exist.

## Introduction

The ventral temporal cortex (VTC) in the brain supports our remarkable ability to recognize objects rapidly and accurately from the visual input in everyday life. Identity information is extracted from visual input through multiple stages of representation. To fully understand the neural mechanism of object processing, it is critical to know how these representations are physically applied to anatomical neural structure in the VTC. Numerous studies have already revealed multiple levels of feature representation manifest at different scales of anatomical organizations which superimposed in the VTC. The superordinate category representations (e.g. animate/inanimate, real-world size) manifest at large scale organization covering the whole VTC. Meanwhile, the category-selective representations (e.g. face, body, and scene selective regions in the mid-fusiform gyrus) are revealed at finer spatial scale in the VTC (*Hasson et al., 2003*; *Spiridon et al., 2006*). Recent evidence suggested a general spatial organization of neural responses to dimensions in object feature space in monkey inferotemporal cortex (*Bao et al., 2020*). Could such physical organization be further extended to even smaller scale, like object parts/features representations within each category-selective region? In other words, as

part representations play a critical role in object processing, would there be consistent spatial tuning across individuals for different object parts within each category-selective region in VTC?

Fine-scale spatial organizations of low-level visual features have already been found in early visual cortex, such as ocular dominance columns and orientation pinwheels (*Blasdel and Salama, 1986*; *Bonhoeffer and Grinvald, 1991*; *Hubel et al., 1977*; *Weliky et al., 1996*). Among all the object-selective regions in the VTC, the face-selective regions, including FFA and OFA, are one of the most widely examined object-processing networks in the past decades in cognitive neuroscience. As faces have spatially separated yet organized features such as eyes and mouth which are easy to be defined, it is suitable to use face parts to examine whether there are spatial tunings for different object features in the VTC. Neurophysiology studies in non-human primates demonstrated face-selective neurons in face-selective regions showed different sensitivities to various of face feature or combination of dimensions in face feature space (*Chang and Tsao, 2017*; *Freiwald et al., 2009*). Human fMRI studies also found the neural response patterns in FFA or OFA could distinguish different face parts (*Zhang et al., 2015*), suggesting voxels within same face-selective region may have different face feature tuning. In addition, previous study also suggests that the spatial distribution of a face feature may be relevant to the physical location of that feature in a face (*Henriksson et al., 2015*).

The sizes of the face-selective regions in VTC are relatively small, spanning about 1 cm. To investigate the potential spatial tuning within each face region, high-resolution fMRI with sufficient sensitivity and spatial precision is necessary. With high-field fMRI, fine-scale patterns have been observed in early visual cortex, such as columnar-like structures in V1, V2, V3, V3a, and hMT (*Cheng et al., 2001*; *Goncalves et al., 2015*; *Nasr et al., 2016*; *Schneider et al., 2019*; *Yacoub et al., 2008*; *Zimmermann et al., 2011*). These findings validate the feasibility of using high-field fMRI to reveal fine-scale (several mm) structures in the visual cortex.

Here, we used 7T fMRI to examine whether category-specific feature information, such as object parts, would be represented in certain spatial pattern within object selective regions. With faces as stimuli, the high-field fMRI allowed for measuring detailed neural response patterns from multiple face-selective regions. Our results show that in the right pFFA and right OFA, different face parts elicited differential spatial patterns of fMRI responses. Specifically, eyes induced responses biased to the posterior portion of the ROIs while responses to mouth and chin were biased to the anterior portion of the ROIs. Similar spatial tuning was observed in both the pFFA and OFA, and the patterns are highly consistent across participants. Together, these results reveal robust fine-scale spatial tuning of face features within face-selective regions.

## Results

One critical challenge to demonstrate the spatial tuning within single face-selective region is to find the anatomical landmark to align the function maps between different individuals, as the shape, size, and spatial location of FFA vary largely across individuals. Among all the anatomical structures in the VTC, the mid-fusiform sulcus (MFS) could potentially serve as landmark in the current study. MFS is relatively small structure in the VTC, but consistently present in most individuals (*Weiner et al., 2014*). On the one hand, the structure of MFS could predict the coordinates of face-selective region around mid-fusiform, especially the anterior one (*Weiner et al., 2014*). On the other hand, MFS is found to be highly consistent with many anatomical lateral-medial transitions in the VTC, such as cytoarchitecture and white-matter connectivity transitions (*Weiner et al., 2014*; *Caspers et al., 2013*; *Grill-Spector and Weiner, 2014*; *Lorenz et al., 2017*). In addition, it could also predict the transitions in many function organization, such as animacy/inanimacy and face/scene preference (*Grill-Spector and Weiner, 2014*). Considering its anatomical and functional significance, in the current study, we used the direction of MFS to align the potential spatial tuning of face part across individuals.

Different face parts (i.e., eyes, nose, mouth, hair, and chin, see *Figure 1A*) and whole faces were presented to participants and they performed a one-back task in 7T MRI scanner. For each participant, five face-selective ROIs (i.e. right pFFA, right aFFA, right OFA, left FFA, and left OFA) were defined with independent localizer scans. Before comparing the spatial response patterns between the face parts, we assessed the overall neural response amplitudes they generated in each ROIs. All face selective regions showed a similar trend that eyes generated higher responses than nose, hair, and chin (ts > 2.61, ps <0.05; except for eyes vs. nose in the left FFA and for eyes vs. chin in left OFA, ts < 2.40,

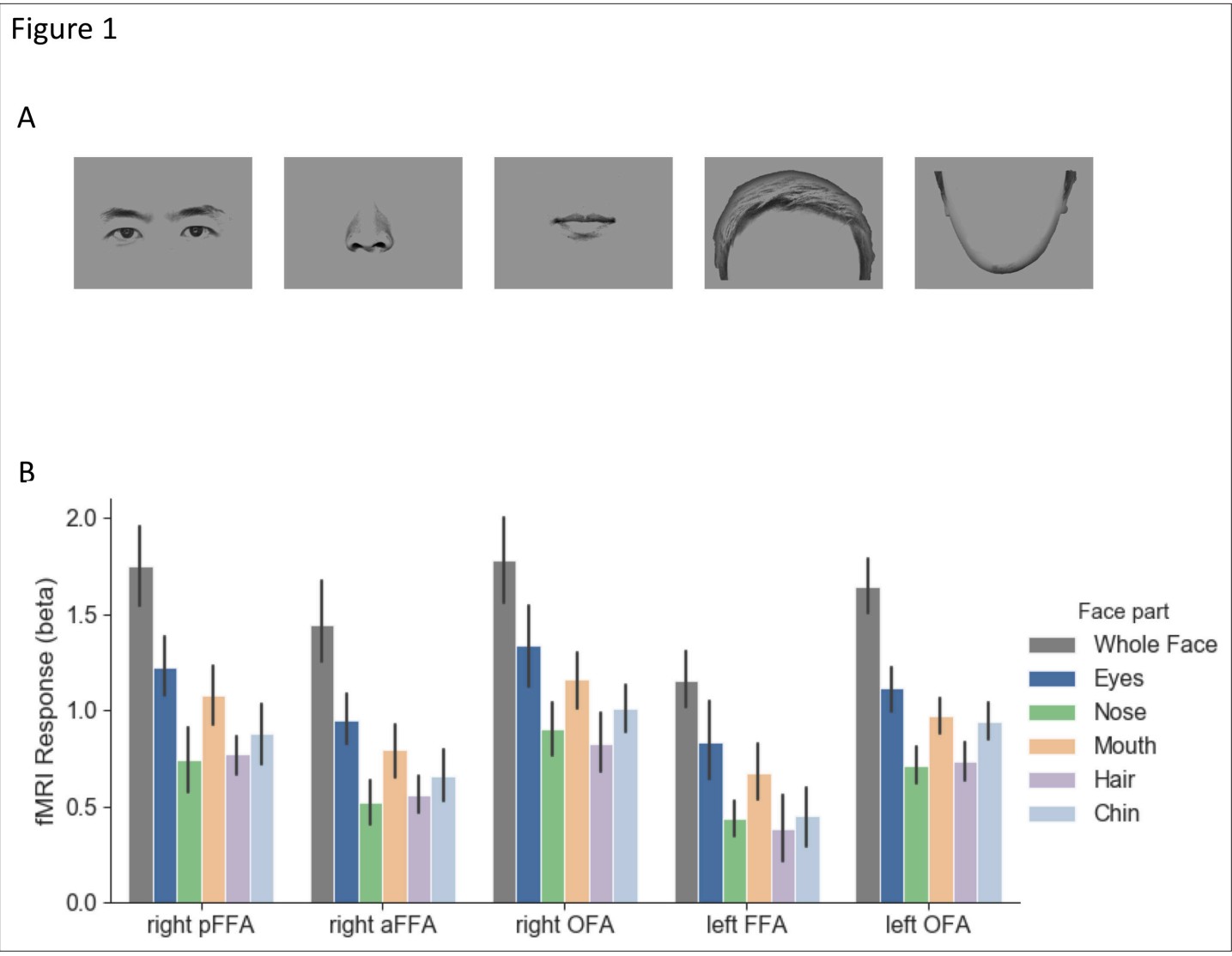

**Figure 1.** Stimuli and fMRI response maps. (**A**) Exemplars of face part stimuli used in the main experiment. The face parts were generated from 20 male faces. Each stimulus was presented around the fixation and participants performed a one-back task during the scan. (**B**) Average fMRI responses to different face parts in each face-selective region. Generally, eyes elicited higher responses than responses to nose, hair, and chin in most of the regions. No significant difference was observed between eyes and mouth responses. Error bars reflect ±1 SEM.

ps >0.06. See *Figure 1B*). However, mouth generated similar response amplitudes as eyes (ts < 1.58, ps >0.17).

Considering that eyes and mouth are two dominant features in face perception (*Schyns et al., 2002*; *Wegrzyn et al., 2017*), and their response amplitudes were similar in face-selective regions, in the initial step, we compared the spatial patterns of neural responses to eyes with that to mouth within each ROI. Each pattern was first normalized to remove any overall amplitude difference between conditions. Then we directly contrasted the two patterns and projected the difference onto the inflated brain surface. A spatial pattern was observed in the right pFFA consistently across all participants (*Figure 2*). In the dimension parallel to the mid-fusiform gyrus, the posterior portion of the right pFFA was biased to respond more to eyes, whereas the anterior portion was biased to respond more to mouth. Note that in participant S2, the direction of MFS was more lateral-medial near the position of the right pFFA, and interestingly, the eyes-mouth contrast map was oriented in the same direction, even though S2's map may initially appear oriented differently from that of other participants. It suggests the anatomical orientation of MFS is highly correlated with such spatial tuning of face parts. To estimate the reliability of such spatial tuning, we split the eight runs data from each

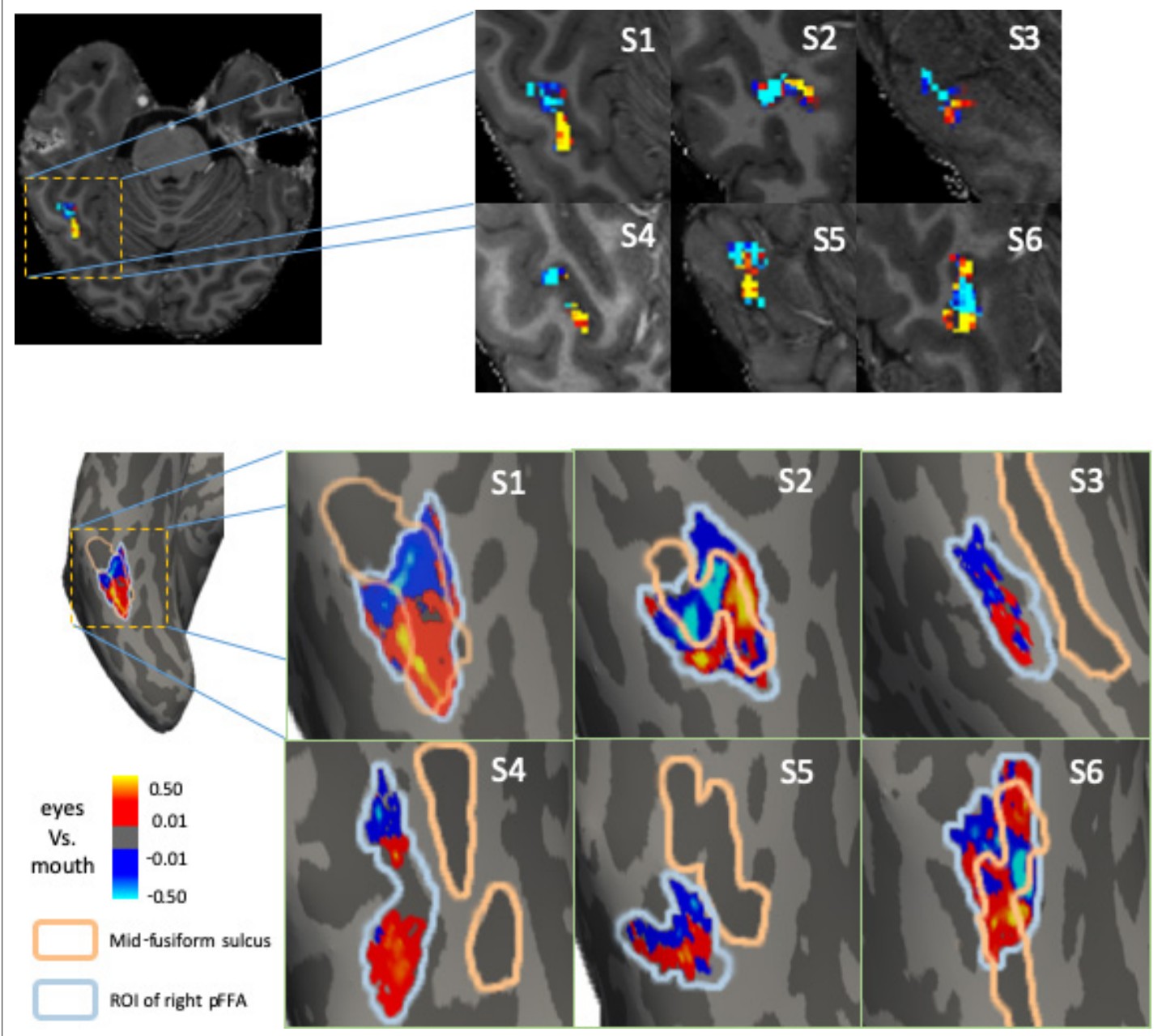

**Figure 2.** Contrast maps between normalized fMRI responses to eyes and mouth in the right pFFA illustrated in the volume (upper) or on inflated cortical surface (lower) of each participant. On the surface, the mid-fusiform sulcus is shown in dark gray with orange outline. The blue line outlines the right pFFA identified with an independent localizer scan. Aligned with the direction of mid-fusiform sulcus, the posterior part of right pFFA shows response bias to eyes (warm colors), while the anterior part illustrates mouth bias (cool colors). The posterior to anterior pattern is generally consistent across participants.

The online version of this article includes the following figure supplement(s) for figure 2:

**Figure supplement 1.** Correlation of eyes-mouth bias across voxels between split half data sets in the main experiment.

participant in the main experiment into two data sets (odd-runs and even-runs), and estimated the eyes-mouth biases within each data set. Then we calculated the correlation coefficient between such biases across different voxels between the two data sets to estimate the reliability of the results in the right pFFA. The results demonstrate strong reliability of the data within participants (*Figure 2—figure supplement 1*).

To further demonstrate such relationship, and also to provide a quantitative description of the spatial tuning of face parts within right pFFA, the fMRI responses to different face parts were projected onto the brain surface of each individual participant. Then we grouped vertices based on their location along the direction parallel to the MFS, and averaged the fMRI responses at each location to generate the response profile on this posterior-anterior dimension (*Figure 3A*, see details in Materials and methods). The group-averaged results clearly showed that the difference between eyes and mouth signals consistently changed along the posterior-anterior direction in the right pFFA (*Figure 3B*). To quantify this trend, we further calculated the correlation coefficient between the eyes-mouth neural response differences and the position index along the posterior-anterior dimension (i.e. more posterior location was assigned with smaller value) in each participant. The group result revealed a significant negative correlation (t(5)=8.36, p = 0.0004, Cohen's d = 3.41), confirming the consistency across participants that the posterior part of right pFFA was biased to eyes and anterior part was biased to mouth.

The contrast map highlighted the differences between eyes and mouth responses. However, the original response patterns elicited by eyes and mouth share the same underlying general 'face-related' pattern, which was subtracted out when contrasting the two response patterns. To extract the response profile of individual face parts, we used independently obtained response patterns of whole faces as the general face-related pattern and regressed it out from the eyes and mouth response patterns. The fMRI responses could be influenced by multiple factors other than neural responses, such as the distribution of the vein, which means there is a shared factor driving the raw fMRI response patterns of different conditions. Thus, to eliminate such shared pattern from the patterns of different face parts, we regressed out the spatial patterns of the whole faces from patterns of each face part. With the general pattern regressed out, we observed distinct spatial profiles elicited by eyes and mouth in the right pFFA (*Figure 3D* top panel). The eye-biased voxels were more posterior than that of mouth-biased voxels, which is consistent with the contrast map shown in *Figure 2*.

Removing the general pattern helped to reveal the pattern of voxel biases for individual face parts. While removing the face-related general pattern achieved this goal, it is possible that removing the general face-related pattern distorted the parts generated response patterns since they share high-level visual information (i.e. face and eyes stimuli are both face-related). Therefore, it is important to check whether the parts specific patterns could be seen with removal of a common face-independent signal distribution. In five of the six participants, data were also obtained when they viewed everyday objects. Indeed, non-face objects generated significantly lower but spatially similar patterns of activation compared with faces across the right pFFA (*Figure 3C*). This result suggests that there is a general intrinsic BOLD sensitivity profile in the pFFA regardless of the stimuli. Indeed, both face and non-face object patterns explained large part of the variation of the face part patterns (for faces average $R^2$ = 0.86, for objects average $R^2$ = 0.72). Thus we proceeded to use the response patterns of either faces or non-face everyday objects to regress out the intrinsic baseline profile from eyes and mouth response patterns, and plotted face part specific patterns along the posterior-anterior dimension. Consistently, results with object patterns removed showed clear posterior bias for eyes and anterior bias for mouth in the right pFFA (*Figure 3D* bottom panel).

To control for the potential contribution from retinotopic bias of the different face part conditions, in our experiment, all stimuli were presented at the fixation with a 1.3° horizontal jitter either to the left or to the right alternatively in different trials within a block. Even though the stimuli were centered on the fixation, because of the nature of the face parts (e.g. two eyes are apart, chin depicts the outline of the face), there were still small degrees (less than 3°) of retinotopic differences between the eyes and mouth conditions. To further rule out the retinotopic contribution, as well as to replicate our finding, we did two control experiments. In the first control experiment (Control Experiment 1), data were obtained with a single eye or mouth presented at either the near central (1.3°) or near peripheral (3.1°) location during the scan (see *Figure 3—figure supplement 1A*). This 2 × 2 (face parts x location) design allowed us to contrasted fMRI response patterns between face parts (single eye vs. mouth) regardless the stimulus location, or between locations (near central vs. near peripheral) regardless the face parts presented. Data from six participants were collected in the Control Experiment 1 and two of them (S1 and S5) also participated main experiment. In all participants, the eye vs. mouth contrast revealed spatial patterns in the right pFFA very similar to that in the main experiment (*Figure 3—figure supplement 1B*). However, contrasting fMRI responses between the near central and near

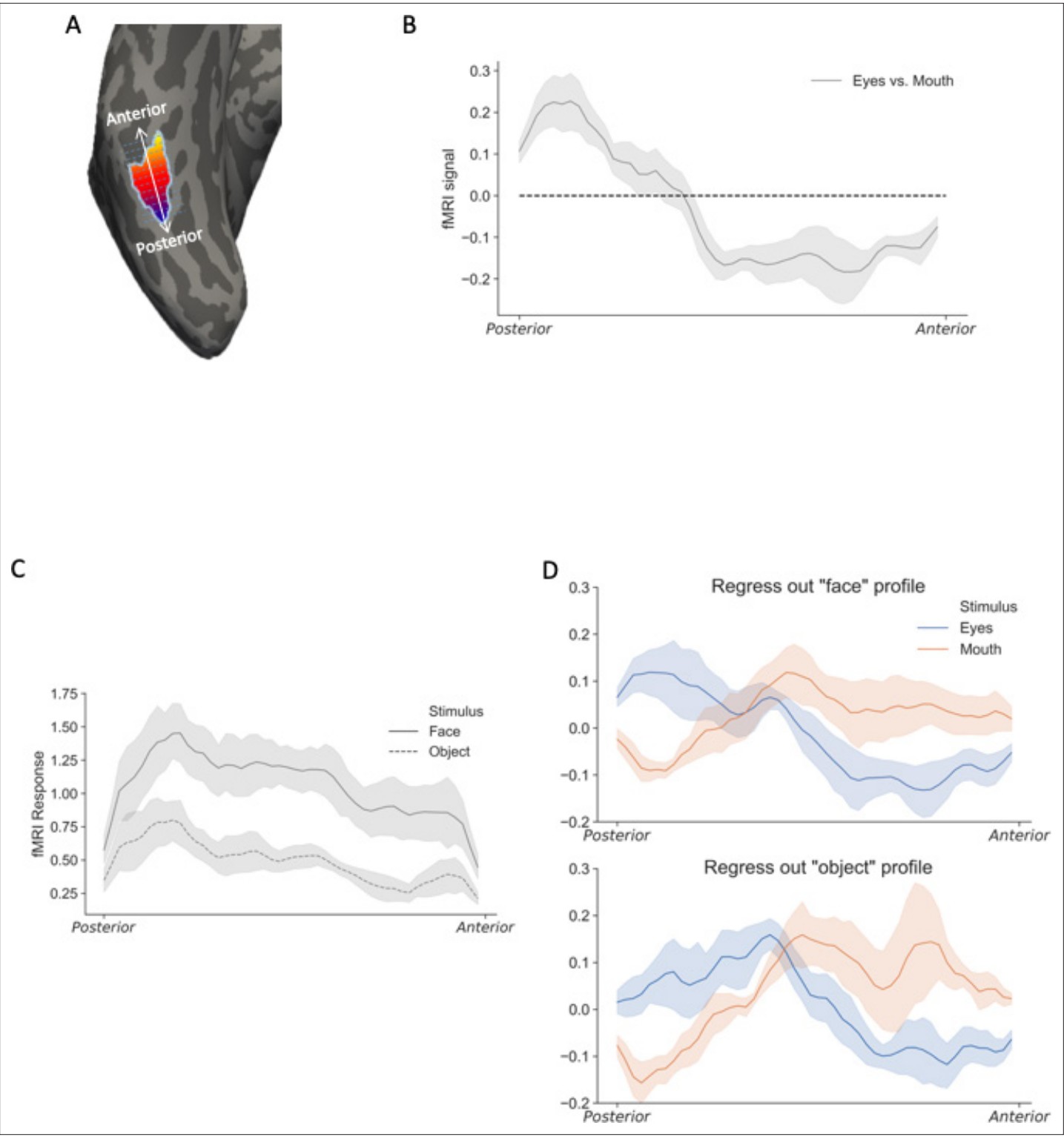

**Figure 3.** The spatial profiles of eyes and mouth responses along the posterior-anterior dimension. (**A**) To obtain the one-dimensional spatial profile of fMRI responses, a line was drawn parallel to the direction of mid-fusiform sulcus. Response from each vertex in the right pFFA was projected to the closest point on the line and averaged to generate the response profile. (**B**) The response profile of eyes vs mouth on the anterior-posterior dimension in right pFFA. The shaded regions reflect ±1 SEM. (**C**) The spatial profiles of whole faces and everyday objects in the right pFFA. Both profiles showed similar patterns, though the whole face responses were generally higher than object responses. (**D**) The spatial profile of individual face part responses, after regressing out the general fMRI response patterns elicited by either the whole faces (upper) or everyday objects (lower). In both cases, distinct spatial profiles were observed between eyes and mouth in the right pFFA.

*Figure 3 continued on next page*

*Figure 3 continued*

The online version of this article includes the following figure supplement(s) for figure 3:

**Figure supplement 1.** Stimuli and results of Control Experiment 1.

**Figure supplement 2.** Stimuli and results of Control Experiment 2.

**Figure supplement 3.** Stimuli and results of pRF experiment.

peripheral location regardless the face parts failed to reveal consistent patterns across participants (*Figure 3—figure supplement 1C*). These results further support that the different fMRI response patterns we observed in the right pFFA were contributed by face feature differences rather than retinotopic bias. In the second control experiment (Control Experiment 2), we used top and bottom parts of the face as stimuli and counterbalance the stimulus location to verify the spatial tuning in the right pFFA. With a 2 × 2 design (eyes vs. nose & mouth x present above vs. below fixation) (*Figure 3—figure supplement 2A*), consistent anterior-posterior spatial patterns in the right pFFA were observed in eight participants (*Figure 3—figure supplement 2B*), which further corroborated our main finding of spatially organized representation of face parts in the right pFFA.

In addition to the two control experiments, we also measured the population receptive field (pRF) of each voxel in the right pFFA in three participants from the main experiment (*Figure 3—figure supplement 3A*) following established procedures (*Dumoulin and Wandell, 2008*; *Kay et al., 2013*; *Kriegeskorte et al., 2008*). For each voxel, parameter x and y were calculated along with other parameters to represent the receptive field center on the horizontal (x) and vertical (y) axis in the visual field. Although generally more voxels in the right pFFA were bias to left visual field, which is consistent with previous report (*Kay et al., 2015*; *Nichols et al., 2016*), we observed no consistent spatial pattern in either x or y map of the right pFFA across participants (*Figure 3—figure supplement 3B*).

To examine the spatial patterns of response from eyes and mouth in other face-selectivity regions, similar analyses as in pFFA were applied to the fMRI response patterns in the right OFA, right aFFA, left FFA, and left OFA. For the left and right OFA, the posterior-anterior dimension was defined as the direction parallel to the occipitotemporal sulcus (OTS), where the OFAs were located in most participants. Among these regions, the right OFA also had distinct response patterns for eyes and mouth along the posterior-anterior dimension (*Figure 4*), similar to what we observed in the right pFFA. Group negative correlation was observed between the eyes-mouth differences and the posterior-anterior location of the right OFA (t(5)=3.63, p = 0.015, Cohen's d = 1.48). Such pattern was also observed in the Control Experiments. We also observed similar spatial patterns between eyes-mouth bias and visual field bias in vertical direction (*Figure 4—figure supplement 1*), which is consistent with previous findings in inferior occipital gyrus (*de Haas et al., 2021*). While the right OFA and right pFFA have been considered as sensitive to facial components and whole faces respectively, in our data they showed similar spatial profiles of eyes and mouth responses along the posterior-anterior dimension. This is consistent with, but adds some constraints to, the idea that the right pFFA may receive face feature information from right OFA for further processing (*Liu et al., 2010*; *Zhu et al., 2011*). In other face-selective regions, no consistent pattern was observed, as the correlations between the eyes-mouth difference and posterior-anterior location were not significant (ts <1.09, ps >0.32, see *Figure 4A*).

Beside the anterior-posterior dimension, the spatial representation of parts could organize in other spatial dimensions, such as the lateral-medial dimension in the VTC, or even in more complex nonlinear patterns. However, since the right pFFA located within the sulcus (MFS) in most of our participants, such that voxels distant from each other on the surface along the lateral-medial dimension could be spatially adjacent in the volume space, making it difficult to accurately reconstruct the spatial pattern along the lateral-medial dimension within the sulcus. Nevertheless, the finding of anterior-posterior bias of face parts is sufficient to demonstrate the existence of fine-scale feature map within object-selective regions.

Our stimuli also included nose, hair, and chin images, thus gave us a chance to examine their spatial profiles in each face-selective ROI as we did for eyes and mouth, though their neural responses were generally lower than that from eyes and mouth. Chin and mouth elicited similar response patterns along the anterior-posterior dimension in the right pFFA and right OFA after regressing out general spatial patterns (*Figure 5A*). By directly contrasting fMRI response patterns between eyes and chin,

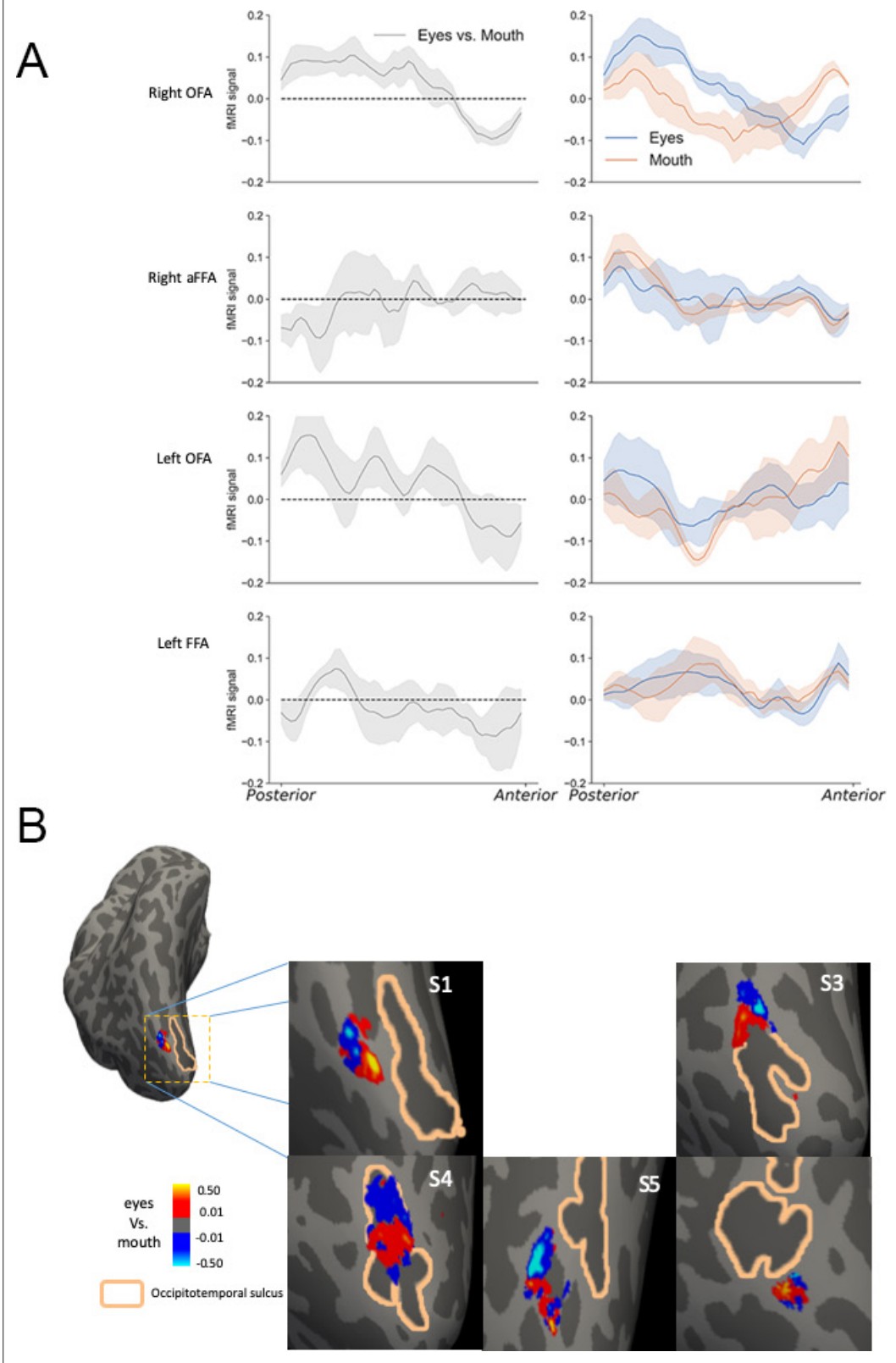

**Figure 4.** The spatial tuning of face parts in other face-selective regions. (**A**) The anterior-posterior neural response profiles of eyes and mouth in other face-selective regions. The contrast between normalized eyes and mouth response patterns in different regions are shown in the left column. The right column plots show the eyes and mouth response profiles with general response patterns regressed out. The right OFA (top panel) demonstrates

*Figure 4 continued on next page*

*Figure 4 continued*

different response profiles for eyes and mouth, similar to the observation in right pFFA. The shaded regions reflect ±1 SEM. (**B**) The eyes vs mouth contrast maps in right OFA in the control experiment 1. A consistent anterior-posterior pattern could be observed in each participant. Right OFA could not be identified in one of the six participants.

The online version of this article includes the following figure supplement(s) for figure 4:

**Figure supplement 1.** Maps of eyes-mouth bias (left column) and vertical visual field bias (right column) in the right OFA.

similar spatial profiles were revealed in the right pFFA and right OFA that the posterior part was biased to eyes and anterior part was biased to chin (ts >5.30, ps <0.01, see *Figure 5B*). We also observed a similar though less obvious profile in the left FFA (t(5)=2.68, p = 0.04, Cohen's d = 1.09), but not in the right aFFA or left OFA (ts <0.41, ps >0.71).

## Discussion

Our results reveal that within certain face-selective regions in the human occipito-temporal cortex, the neural representations of different facial features have consistent spatial profiles. Such fine-scale spatial tuning is found similarly in the right pFFA and right OFA, but not in the more anterior right aFFA nor in the left hemisphere's face-selective regions. In other words, fine-scale spatial tuning for face parts exists at the early to intermediate stages of face processing hierarchy in the right hemisphere.

In the current study, five face parts (i.e. eyes, nose, mouth, hair, and chin) were tested, with eyes and mouth showed most distinct spatial profiles in the right pFFA and right OFA. No obvious spatial pattern was observed for nose and hair in face-selective regions, but it would be premature to conclude that there is no fine-scale spatial profile for their neural representations. For one, the nose and hair stimuli elicited lower fMRI responses compared with eyes and mouth stimuli, making it more difficult to detect potential spatial patterns. The observation that eyes and mouth elicited most differential patterns is consistent with them providing more information about faces than other features in face processing (*Schyns et al., 2002*; *Wegrzyn et al., 2017*). The dominance of eyes and mouth in face-selective regions could be considered as a form of cortical magnification of more informative features, a common principle of functional organization in sensory cortex (*Cowey and Rolls, 1974*; *Daniel and Whitteridge, 1961*; *Penfield and Boldrey, 1937*).

The discovery that some face parts are represented within the face-processing regions with fine-scale spatial tuning improve our understanding about how functional representations are physically applied to anatomical structures in the VTC. To further explore the neural models about object processing in the VTC, it is important to ask what kinds of constrain, functional or anatomical, result in such fine-scale spatial tuning? Many of the visual cortical areas have retinotopic maps, indeed having a retinotopic representation of the visual world is one of the key ways to define a visual cortical area. Along occipitotemporal processing stream, visual areas increasingly become more specialized in processing certain features and object categories. What is the relationship between a potential spatial organization of face part representations and the spatial relationship of face parts in a real face?

A recent study has revealed that in the inferior occipital gyrus, where the OFA located, both tunings for retinotopic location and face parts (*de Haas et al., 2021*). Although the tuning peak maps were idiosyncratic across individuals, the two tuning maps were correlated within individuals, suggesting a relationship between face parts configuration and their typical retinotopic configuration. Our findings provide additional support for face part turning in the OFA, and further reveal that there exists a consistent spatial profile of face part tuning *across* individuals. More importantly, our finding of spatial tuning of face part in the pFFA indicates that although the organization of feature tuning could be constrained by the retinotopic tuning in occipital cortex, a more abstract feature tuning could still be spatially organized in cortical areas with absent or minimal retinotopic property in the later stages of VTC.

Another previous study also tested the idea of 'faciotopy', that there are cortical patches representing different face features within a face-selective region and the spatial organization of these feature patches on the cortical surface would reflect the physical relationships of face features (*Henriksson et al., 2015*). Their results showed that in the OFA and FFA, the differences between

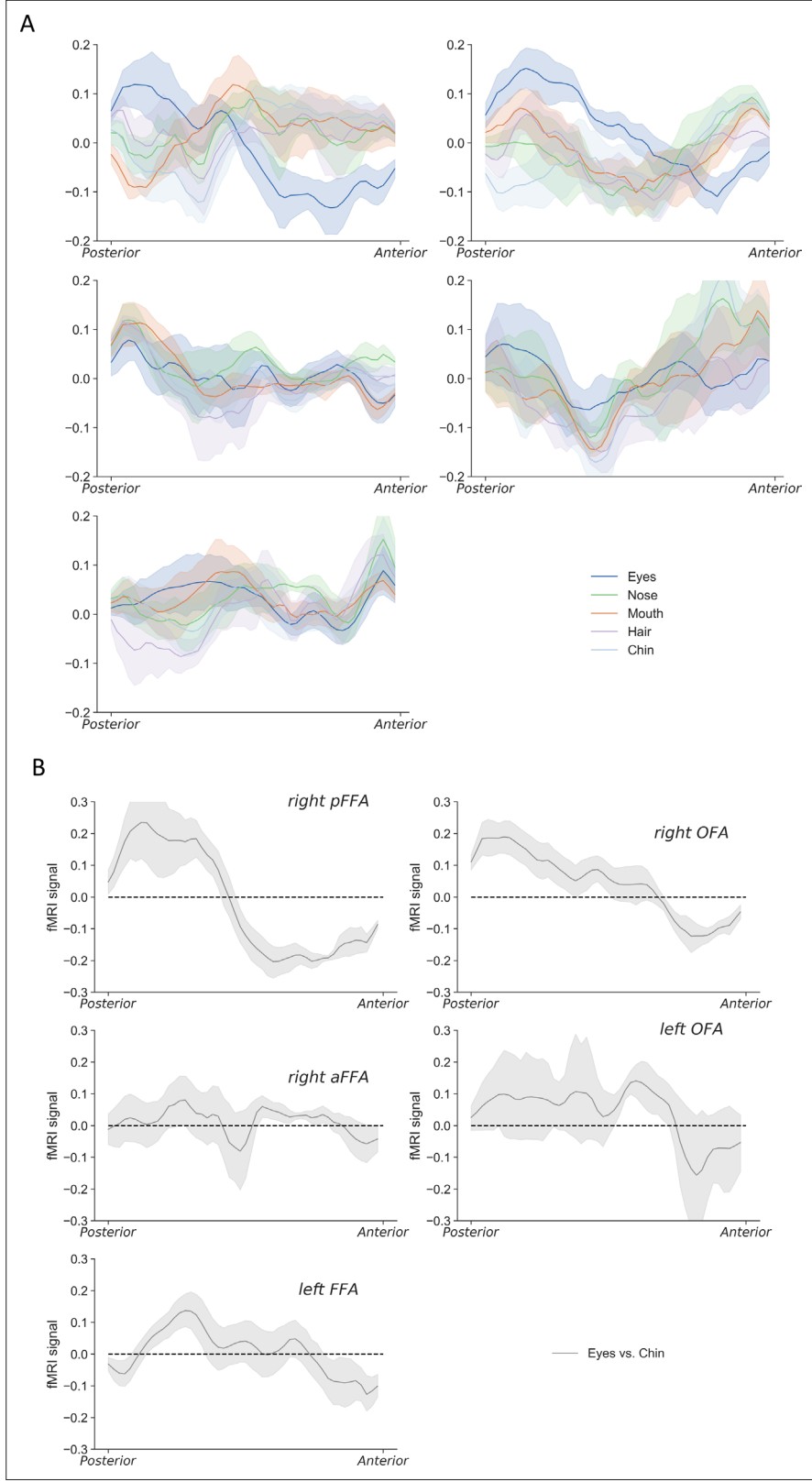

**Figure 5.** The anterior-posterior neural response profiles of all face parts (i.e. eyes, nose, mouth, hair, and chin) in five face-selective regions. (**A**) With general pattern regressed out, the chin showed similar response profiles as mouth in right pFFA and right OFA. (**B**) Contrasting normalized eyes and chin response patterns revealed consistent changes along the posterior-anterior dimension in right pFFA and right OFA. The shaded regions reflect ±1 SEM.

neural response patterns of face parts were correlated with physical distances between face parts in a face. Our results support the existence of stable spatial profile of face features in the right OFA and right pFFA, especially for eyes and mouth. The possible mechanism underlying such faciotopy organization is the local-to-global computation, that physically adjacent face parts interact more than parts far apart from each other during the early stages of processing, thus it is more efficient for them to have neural representations near each other. However, in the current study, we did not find the posterior bias pattern for hair as we did for eyes, even though hair and eyes are spatially adjacent, which could be caused by the hair being generally less invariant and less informative in the face identification.

Another potential explanation could be that while both contribute to face identification, eyes and mouth are differentially involved in different neural networks and have distinct functional connectivity profiles with other brain regions. Specifically, the mouth region provides more information for language processing and audio-visual communication perception, thus it may be more connected to the language processing system. Meanwhile, the eyes are more important in face detection and eye gaze signifies interest, thus it may be more connected to the attention system. Previous studies have already found the connectivity profiles could predict the functional selectivity in the VTC, thus it would be interesting to examine whether the face part spatial tuning in the pFFA could be predicted using functional or anatomical connectivity to the other brain regions in the future studies.

The third explanation of our results is that the fine-scale pattern of face part sensitivity is driven by larger-scale organization of object-selective regions in the ventral pathway. As the FFA is overlapped with body-selective region fusiform-body area (FBA), it is possible that some face features (e.g. mouth) could be represented closer to the FBA, while face parts such as the eyes are more represented in the FFA proper. However, existing evidence does not support a consistent anterior-posterior relationship between FFA and FBA (*Kim et al., 2014*). It remains important to directly compare the eyes-mouth pattern against the face-body pattern with high-resolution fMRI in future studies.

The spatial clustering of neurons with similar tuning is one of the organization principles in the brain. Such clustering may optimize the efficiency of the neural computation by reducing the total wire length (*Laughlin and Sejnowski, 2003*), thus the clustering of neurons with similar feature tuning within face-selective regions could improve the processing efficiency to face stimuli. Our results provide evidence from neural imaging data to support the voxel level neuronal clustering driven by the tuning to different face parts. Previous fMRI and single unit recording studies in monkey face processing network have demonstrated strong correspondence between fMRI signal and neuronal responses (*Tsao et al., 2006*), suggesting that the face part tuning in our results may be driven by neuronal response biases. In addition to neuronal response biases, the clustering could also reflect activity synchronization across neurons. Further neurophysiology studies are needed to delineate the specific mechanisms to the spatial clustering observed within face-selective regions.

Among five face-selective regions we examined, only the right pFFA and right OFA exhibited distinct fMRI response patterns for eyes and mouth. In the face processing network, face parts are believed to be represented in the posterior regions such as the OFA (*Liu et al., 2010*; *Arcurio et al., 2012*; *Pitcher et al., 2007*). Part information is transmitted to anterior regions to be further integrated to form holistic face representations (*Zhang et al., 2015*; *Rotshtein et al., 2005*). In that sense, the more anterior regions in the face processing network are more responsible for representing integrated face information such as gender or identity rather than individual face parts (*Freiwald and Tsao, 2010*; *Landi and Freiwald, 2017*). Consistent with this idea, at the right aFFA, there is no obvious spatial tuning of face parts. Meanwhile, a clear hemispheric difference was found in our results that the distinct spatial response patterns for face parts were observed in the right but not the left hemisphere, which is consistent with previous findings that compared with left FFA, right FFA is more sensitive to face specific features (*Meng et al., 2012*) and configural information (*Rossion et al., 2000*). The neural clustering of face part tuning and consistent spatial patterns across individuals in the right rather than in the left face selective regions provides a potential computational advantage for right lateralization for face processing. The clustering of neurons with similar feature tuning have been found extensively in the ventral pathway, which may help to support a more efficient neural processing. Therefore, one of the neural mechanisms underlying the functional lateralization of face processing could be the existence of spatial clustering of face part tunings in the right hemisphere.

Much progress has been made in our understanding of object feature representation in the VTC during the past decade, especially with the view of feature space representation (*Bao et al., 2020*; *Chang and Tsao, 2017*). Consequently, we now believe that a large number of features are represented for object recognition, but how does our brain physically organize such complex feature representations in the VTC? One possible solution is that these feature representations manifest in different spatial scales. For more general features the representation manifests at large spatial scale across the whole VTC (e.g. large/small, animate/inanimate), and for more specific features such as face parts, it manifests at finer spatial scales within specific object processing regions. Under this view, we would expect more fine-scale feature organizations to be revealed with more advanced neural imaging tools, which are critical for fully understanding the neural algorithm of object processing in the VTC.

## Materials and methods
### Participants
Six (3 females) human participants were recruited in the main experiment. Six (5 females) participants (two of them also participated main experiment) were recruited in the Control Experiment 1. Three participants (2 females) from main experiment finished the pRF experiment. Ten participants (one female) were recruited in the Control Experiment 2, but in two participants right pFFA was failed to be localized, thus we excluded these two participants from the analyses. All participants were between the ages of 21 and 27, right-handed, and had normal or corrected to normal visual acuity. They were recruited from the Chinese Academy of Sciences community with informed consent and received payment for their participation in the experiment. The experiment was approved by the Committee on the Use of Human Subjects at the Institute of Biophysics of Chinese Academy of Sciences (#2017-IRB-004).

### Stimuli and experimental design
In the main experiment, for face stimuli, 20 unique front-view Asian male face images were used. Each face image was gray-scaled and further divided into five parts (i.e. eyes, nose, mouth, hair, and chin. See *Figure 1A*). Twenty unique gray-scaled everyday objects were used as comparison stimuli. The full face and object images on average subtended around 5° x 7°. For stimuli used in localizer scans, video clips of faces, objects, and scrambled objects were used (For detail see *Pitcher et al., 2011*).

There were total of seven stimulus conditions (i.e. eyes, nose, mouth, hair, chin, whole face, and object condition). Each main experimental run contained two blocks of each stimulus condition. In the scan of participant S6, the object condition was not included. Each stimulus block lasted 16 s and contained 20 images of the same type. Each image was presented for 600 ms at fixation and followed by a 200 msec blank interval. There was a 16 s blank fixation block at the beginning, the middle, and the end of each run. Participants performed a 1-back task that they were asked to press a button when two successive images were the same. To balance the spatial property in the visual field of different images, each image was presented at a slightly shifted location, 1.3° either to the left or to the right of the fixation alternately in different trials within a block. Participants were instructed to maintain central fixation throughout the task.

Each localizer run contained four 18 s blocks of each of the three stimulus conditions (i.e. faces, everyday non-face objects, and scrambled objects) shown in a balanced block order. The 12 stimulus blocks were interleaved by three 18 s fixation blocks inserted at the beginning, middle and end of each run. Each block contained six video clips of a given stimulus category, each presented for 3 s. Participants were asked to watch the video without any task. No fixation point was presented during the scan.

The eight experimental runs and the two localizer runs were completed within the same scan session for each participant.

In the Control Experiment 1, we used a similar block design as that in the main experiment. There were six kinds of stimulus blocks (single eye near central, single eye near peripheral, mouth near central, mouth near peripheral, whole face, object) and each of them repeated three times in a single run. Each participant completed four runs and two localizer runs. In the eye near central condition, single left eye images were presented at 1.3° either to the left or to the right of the fixation alternately in different trials within a block. In the eye near peripheral condition, single left eye images were

presented at 3.1° either to the left or to the right of the fixation. The central and peripheral locations were chosen to match the locations of eyes and mouth in the main experiment. Stimuli in mouth near central and mouth near peripheral conditions were presented in the same locations as in two eye conditions, respectively. Whole face and object conditions were the same as in the main experiment.

In the pRF experiment, we adopted stimuli and analysis code from analyzePRF package (http://kendrickkay.net/analyzePRF/). There were total of four conditions (i.e. clockwise wedges, counter-clockwise wedges, expanding rings, contracting rings). The angular span of the wedges was 45°, and it revolved for 32 s per cycle. In the rings conditions, the rings swept 28 s per cycle with 4 s of rest followed. Colored object images were presented on the wedges or rings. The rings and wedges were presented within a radius of 10°. For each run, there was a 22 s blank fixation block at the beginning and the end. Participants performed a change detection task that they pressed a button whenever the fixation color changed. In each run, only one kind of PRF stimulus was presented and repeated eight cycles. Each participant finished four different pRF runs.

In the Control Experiment 2, similar block-design as in main experiment was used. Four face part conditions (top vs bottom part x present location) were included in the experiment (*Figure 3—figure supplement 2A*). The top part contained eyes (4.02° x 12.08°) and the bottom part contained nose and mouth (8.08° x 12.08°). To engage observers' attention on the stimuli, a randomly selected four images in each block moved slightly either to the left or right during stimulus presentation. Observers were asked to judge the directions of these movements. Same localizer runs as in the main experiment were included for each participant.

## FMRI scanning

MRI data were collected on Siemens Magnetom 7 Tesla MRI system (passively shielded, 45mT/s slew rate) (Siemens, Erlangen, Germany), with a 32-channel receive 1-channel transmit head coil (NOVA Medical, Inc, Wilmington, MA, USA), at the Beijing MRI Center for Brain Research (BMCBR). High-resolution T1-weighted anatomical images (0.7 mm isotropic voxel size) were acquired with a MPRAGE sequence (256 sagittal slices, acquisition matrix = 320 × 320, Field of view = 223 × 223 mm, GRAPPA factor = 3, TR = 3100ms, TE = 3.56ms, TI = 1250ms, flip angle = 5°, pixel bandwidth = 182 Hz per pixel). Proton density (PD)-weighted images were acquired with same voxel size and FOV (256 sagittal slices, acquisition matrix = 320 × 320, Field of view = 223 × 223 mm, GRAPPA factor = 3, TR = 2340ms, TE = 3.56ms, flip angle = 5°, pixel bandwidth = 182 Hz). GE-EPI sequences was used to collect functional data in the main experiment (TR = 2000ms, TE = 18ms, 1.2 mm isotropic voxels, FOV = 168 × 168 mm, image matrix = 140 × 140, GRAPPA factor = 3, partial Fourier 6/8, 31 slices of 1.2 mm thinkness, flip angle is about 80, pixel bandwidth = 1276 Hz per pixel). During the scan, GE-EPI images with reversed phase encoding direction from experiment functional scan were collected to correct the spatial distortion of EPI images (*Morgan et al., 2004*). Dielectric pads were placed on both sides of the head to improve B1 efficiency in the temporal cortex (*Teeuwisse et al., 2012*), while bite-bar was used to reduce head movements for each participant. During the functional scan, respiration and pulse signals were recorded simultaneously. GE-EPI sequences with same resolution as in the main experiment was used in the control and pRF experiment (TR = 2000 ms, TE = 22ms, 1.2 mm isotropic voxels, FOV = 180 × 180 mm, image matrix = 150 × 150, GRAPPA factor = 2, partial Fourier 6/8, 31 slices of 1.2 mm thinkness, flip angle is about 80, pixel bandwidth = 1587 Hz per pixel). Dielectric pads were placed on the right side of the head.

## Data analysis

Anatomical data were analyzed with FreeSurfer (Cortechs Inc, Charlestown, MA) and custom MATLAB codes. To enhance the contrast between white and gray matter, T1-weighted images were divided by PD-weighted images (*Van de Moortele et al., 2009*). Anatomical data were further processed with FreeSurfer to reconstruct the cortical surface models.

Functional data were analyzed with AFNI (http://afni.nimh.nih.gov), FreeSurfer, fROI (http://froi.sourceforge.net), and custom MATLAB codes. Data preprocessing included slice-timing correction, motion correction, removing physiological noise with respiration and pulse signals, distortion correction with reversed phase encoding EPI images, and intensity normalization. For the localizer runs only, spatial smoothing was applied (Gaussian kernel, 2 mm full width at half maximum). After preprocessing, function images were co-registered to anatomic images for each participant. To obtain

the average response amplitude for each voxel in the specific stimulus condition for each individual observer, voxel time courses were fitted by a general linear model (GLM), whereby each condition was modeled by a boxcar regressor (matched in stimulus duration) and then convolved with a gamma function (delta = 2.25, tau = 1.25). The resulting beta weights were used to characterize the averaged response amplitudes.

The face-selective ROIs were identified by contrasting functional data between face and everyday-object conditions in the localizer runs. Specifically, FFA and OFA was defined as the set of continuous voxels in fusiform gyrus and inferior occipital gyrus, respectively, that showed significantly higher response to faces than to objects ($p < 0.01$, uncorrected). We were able to identify right pFFA, right anterior FFA (right aFFA), right OFA, and left FFA in all six participants. The left OFA were successfully identified in five out of six participants. In each ROI, to remove the vein signal in the functional data, voxels of which signal changes to face stimuli were larger than 4% were excluded in further analysis.

For the main experimental data, to remove the general fMRI response pattern shared among different face parts, response patterns from whole faces or everyday objects were regressed out from response patterns of each individual face part. Whole face or object response in each voxel was used to predict the individual part response with linear regression algorithm, and the residuals across voxels were considered as the individual part response pattern with general response pattern removed. To extract the trend of the fMRI response pattern along anterior-posterior dimension in the FFA, we first drew a line along the mid-fusiform sulcus on the cortical surface of each participant. For all vertices within the FFA ROI, we calculated their shortest (orthogonal) distances to the line, and projected the neural response of all voxels in the FFA ROI to the line along the mid-fusiform sulcus, and obtained the averaged response on each point along the line to get the response profiles (see *Figure 3A*). Similar analysis was done for OFA with the line drawn along the inferior occipital sulcus.

For the control experiments, same data processing steps as in the main experiment were applied to extract the spatial patterns of different conditions. For the pRF data, fMRI respond time course of each voxel was fit with compressive spatial summation (CSS) model (http://kendrickkay.net/analyzePRF/). To determine the center location (x, y) of each voxel's population receptive field, CSS used an isotropic 2D Gaussian and a static power-low nonlinearity to model the fMRI response. In each voxel, model fitness can be quantified as the coefficient of determination between model and data ($R^2$). We only included the pRF results of voxels with $R^2$ higher than 2%.

## Acknowledgements

This work was supported by funds from NSFC grant (No. 31800966), CAS Pioneer Hundred Talents Program, Strategy Priority Research Program of Chinese Academy of Science (No. XDB32020200), and CAS Key Research Program of Frontier Sciences (No. KJZD-SW-L08). The authors would like to thank Dr. Chencan Qian and Dr. Zihao Zhang for their help during data collection and analysis.

## Additional information

### Funding

| Funder | Grant reference number | Author |
| --- | --- | --- |
| National Natural Science Foundation of China | 31800966 | Jiedong Zhang |
| Chinese Academy of Sciences | CAS Pioneer Hundred Talents Program | Jiedong Zhang |
| Chinese Academy of Sciences | XDB32020200 | Sheng He |
| Chinese Academy of Sciences | KJZD-SW-L08 | Sheng He |

The funders had no role in study design, data collection and interpretation, or the decision to submit the work for publication.

## Author contributions
Jiedong Zhang, Conceptualization, Data curation, Formal analysis, Funding acquisition, Investigation, Methodology, Project administration, Resources, Software, Supervision, Validation, Visualization, Writing – original draft, Writing – review and editing; Yong Jiang, Data curation, Formal analysis, Methodology, Validation, Visualization; Yunjie Song, Data curation, Formal analysis, Validation, Visualization; Peng Zhang, Data curation, Investigation, Methodology, Validation, Writing – original draft, Writing – review and editing; Sheng He, Funding acquisition, Methodology, Project administration, Supervision, Writing – original draft, Writing – review and editing

## Author ORCIDs
Jiedong Zhang http://orcid.org/0000-0002-4432-2752
Sheng He http://orcid.org/0000-0001-5547-923X

## Ethics
Human subjects: All human participants were recruited from the Chinese Academy of Sciences community with informed consent and received payment for their participation in the experiment. The experiment was approved by the Committee on the Use of Human Subjects at the Institute of Biophysics of Chinese Academy of Sciences (#2017-IRB-004).

## Decision letter and Author response
Decision letter https://doi.org/10.7554/eLife.70925.sa1
Author response https://doi.org/10.7554/eLife.70925.sa2

## Additional files

### Supplementary files
• Transparent reporting form

### Data availability
fMRI response data have been deposited in Dryad.

The following dataset was generated:

| Author(s) | Year | Dataset title | Dataset URL | Database and Identifier |
|---|---|---|---|---|
| Zhang J, Jiang Y, Song Y, Zhang P, He S | 2021 | Dataset for Spatial tuning of face part representations within face-selective areas revealed by high-field fMRI | https://doi.org/10.5061/dryad.gmsbcc2nh | Dryad Digital Repository, 10.5061/dryad.gmsbcc2nh |

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
