## [Editor Report]

How the brain is organized to represent various concepts has long been a central cognitive neuroscience research topic. Zhang and colleagues investigated the spatial distribution of feature tuning for different face-parts within face-selective regions of human visual cortex using ultra-high resolution 7.0 T fMRI. The findings complement non-human primate studies of face-selective patches and will be of interest to psychologists and system neuroscientists.

---

## [Decision Letter]

**Decision letter after peer review:**

Thank you for submitting your article "Spatial organization of face part representations within face-selective areas revealed by high-field fMRI" for consideration by *eLife*. Your article has been reviewed by 3 peer reviewers, one of whom is a member of our Board of Reviewing Editors, and the evaluation has been overseen by and Chris Baker as the Senior Editor. The following individual involved in review of your submission has agreed to reveal their identity: Ed H Silson (Reviewer #3).

Essential revisions:

1) Tune down the claim from functional organization to functional preference. Typically, when we talk about the organization, it either has more than 2 subdivisions or it has a continuous representation about certain features. In this paper, the results are mainly about the comparison between two face parts, and they do not reveal other distinctive subareas showing preference to other face parts.

2) Clarify interpretation/motivation of the results: other than "faciotopy", why would neurons that are tuning to different face parts be spatially clustered? Given that BOLD signals are indirect measurements of neural activity, can the results about fMRI voxels really suggest neuronal clustering? Would alternative interpretations such as activity synchronization be possible?

3) The inclusion of a non-face object condition was nice. However, it might be helpful to clarify the logic behind regressing out response patterns from whole faces or everyday objects. Wouldn't it make more sense to regress out response patterns from the comparison between whole faces and everyday objects, i.e. faces – objects?

4) It is currently unclear whether the current data are in full agreement with recent work (de Haas et al., 2021) showing similar face-part tuning within the OFA (or IOG) bilaterally. The current data suggest that feature tuning for eye and mouth parts progresses along the posterior-anterior axis within the right pFFA and right OFA. In this regard, the data are consistent. But de Haas and colleagues also demonstrated tuning for visual space that was spatially correlated (i.e. upper visual field representations overlapped upper face-part preferences and vice-versa). The current manuscript found little evidence for this correspondence within pFFA but does not report the data for OFA. For completeness this should be reported and any discrepancies with either the prior work, or between OFA and pFFA discussed.

5) It is somewhat challenging to fully interpret the responses to face-parts when they were presented at fixation and not in the typical visual field locations during real-world perception. For instance, we typically fixate faces either on or just below the eyes (Peterson et al., 2012) and so in the current experiment the eyes are in the typical viewing position, but the remainder of the face-parts are not (e.g. when fixating the eyes, the nose mouth and chin all fall in the lower visual field but in the current experimental paradigm they appear at fixation). Consideration of whether the reported face-part tuning would hold (or even be enhanced) if face-parts were presented in their typical locations should be included.

6) Although several experiments (including two controls) have been conducted, each one runs the risk of being underpowered (n ranges 3-10). One way to add reassurance when sample sizes are small is to include analyses of the reliability and replicability of the data within subjects through a split-half, or other cross-validation procedure. The main experiment here consisted of eight functional runs, which is more than sufficient for these types of analyses to be performed.

7) The current findings were only present within the right pFFA and right OFA. Although right lateralisation of face-processing is mentioned in the discussion, this is only cursory. A more expansive discussion of what such a face-part tuning might mean for our understanding of face-processing is warranted, particularly given that the recent work by de Haas and colleagues was bilateral.

*Reviewer #1 (Recommendations for the authors):*

While I think in general this is a well-executed study with powerful high-field 7T fMRI, I still have a few suggestions for the authors to consider.

1. Clarify interpretation/motivation of the results: other than "faciotopy", why would neurons that are tuning to different face parts be spatially clustered? Given that BOLD signals are indirect measurements of neural activity, can the results about fmri voxels really suggest neuronal clustering? Would alternative interpretations such as activity synchronization be possible?

2. The inclusion of a non-face object condition was nice. However, it might be helpful to clarify the logic behind regressing out response patterns from whole faces or everyday objects. Wouldn't it make more sense to regress out response patterns from the comparison between whole faces and everyday objects, i.e. faces – objects?

3. Testing functional connectivity profiles is a very interesting idea. Why not to take a look with the coverage of present data or perhaps datasets from openly available sources?

*Reviewer #3 (Recommendations for the authors):*

1) Can the authors make it clear whether the analyses concerning the P-A axis were conducted across voxels (i.e. volume space) or across surface vertices (i.e. surface space).

2) An uncorrected threshold pf p<0.01 for a functional localiser is non-standard and very liberal. Even with two-runs some form of reliability analysis can be performed.

3) Similarly, the main experiment comprises 8 functional runs. This amount of data can easily be split to perform reliability and reproducibility analyses. These would bolster the main claims as each experiment is relatively underpowered.

4) A pRF threshold of R2 = 0.02 (2%) is again very liberal. Can this selection be justified? Many prior pRF works even within FFA/PPA use a much higher threshold (~10-20%).

---

## [Author Response]

Essential revisions:1) Tune down the claim from functional organization to functional preference. Typically, when we talk about the organization, it either has more than 2 subdivisions or it has a continuous representation about certain features. In this paper, the results are mainly about the comparison between two face parts, and they do not reveal other distinctive subareas showing preference to other face parts.

We have followed the advice from the reviewer to tune down the claim of functional organization in our manuscript. To emphasize both the functional preferences to different face parts within face-selective regions and the consistent spatial profile across different individuals, we now use “spatial tuning of face parts” in the manuscript. The revision has been made in the manuscript.

2) Clarify interpretation/motivation of the results: other than "faciotopy", why would neurons that are tuning to different face parts be spatially clustered? Given that BOLD signals are indirect measurements of neural activity, can the results about fMRI voxels really suggest neuronal clustering? Would alternative interpretations such as activity synchronization be possible?

The spatial clustering of neurons with similar tuning functions is one of the organization principles in the brain. Such clustering may optimize the efficiency of the neural computation by reducing the total wire length (Laughlin and Sejnowski, 2003), and is widely observed in the ventral pathway at different spatial scales (e.g., ocular dominance columns in V1, category-selective areas in VTC). Within category-selective regions like FFA, previous studies have found subpopulations with different response biases (Cukur et al., 2013; Weiner and Grill-Spector, 2010) or different connectivity profiles to other brain regions (Park et al., 2017), thus the clustering of neurons with similar feature tuning within face-selective regions could improve the processing efficiency to face stimuli.

Our results provide evidence from neural imaging data to support the voxel level neuronal clustering driven by the tuning to different face parts. Previous fMRI and single unit recording studies in monkey face processing network have demonstrated strong correspondence between fMRI signal and neuronal responses (Tsao et al., 2006), suggesting the face part tuning in our results may be driven by neuronal response biases. Here the reviewer has pointed out another possibility that the clustering could also reflect activity synchronization. These two mechanisms may interact to generate clustering at different spatial scales. We have added more discussion about this in the discussion session of the manuscript (Page 24).

3) The inclusion of a non-face object condition was nice. However, it might be helpful to clarify the logic behind regressing out response patterns from whole faces or everyday objects. Wouldn't it make more sense to regress out response patterns from the comparison between whole faces and everyday objects, i.e. faces – objects?

Regressing out the face and non-face object response pattern from the face part pattern is an important step in revealing the face part preferences in our results. We thank the reviewer for reminding us to clarify the logic behind this operation. The fMRI responses could be influenced by multiple factors other than neural responses, such as the distribution of the vein. While many studies have showed that the neural representations of faces and non-face objects are quite different in the pFFA, however, as shown in the Figure 3C in the manuscript, their spatial distributions of raw Β value are similar, which means that there is a shared factor driving the raw fMRI response patterns. Thus to eliminate such shared pattern from the patterns of different face parts, we regressed out the spatial patterns of the whole faces from patterns of each face part. Regression rather than subtraction was used here to avoid distorting the patterns specific to face parts. In addition, we regressed out response patterns of non-face objects, which further ensured that there was no distortion to the patterns specific to face parts. Both regression analyses yielded clear and similar response patterns of face parts. We have added more description about the logic behind the regression analysis in the Results session (Page 9).

Based on this logic, the pattern of “faces – objects” may not be a better regressor, as the shared patterns may be largely removed by the subtraction.

4) It is currently unclear whether the current data are in full agreement with recent work (de Haas et al., 2021) showing similar face-part tuning within the OFA (or IOG) bilaterally. The current data suggest that feature tuning for eye and mouth parts progresses along the posterior-anterior axis within the right pFFA and right OFA. In this regard, the data are consistent. But de Haas and colleagues also demonstrated tuning for visual space that was spatially correlated (i.e. upper visual field representations overlapped upper face-part preferences and vice-versa). The current manuscript found little evidence for this correspondence within pFFA but does not report the data for OFA. For completeness this should be reported and any discrepancies with either the prior work, or between OFA and pFFA discussed.

In the current study, three participants had data from both retinotopic mapping and face part mapping experiments. Consistent and robust part clustering were found in the right pFFA and right OFA. Following the reviewer’s suggestion, we analyzed these data for the right OFA and found the spatial patterns of eyes vs. mouths are similar to the patterns of visual field sensitivity on the vertical direction (i.e., upper to lower visual field), which are consistent with de Haas and colleagues’ findings. Note that we used more precise functional localization of OFA, while de Haas et al’s analysis was based on anatomically defined IOG, for which OFA is a part of. We have added this result in the Results session (Page 16), and also added a supplemental Figure 4—figure supplement 1.

5) It is somewhat challenging to fully interpret the responses to face-parts when they were presented at fixation and not in the typical visual field locations during real-world perception. For instance, we typically fixate faces either on or just below the eyes (Peterson et al., 2012) and so in the current experiment the eyes are in the typical viewing position, but the remainder of the face-parts are not (e.g. when fixating the eyes, the nose mouth and chin all fall in the lower visual field but in the current experimental paradigm they appear at fixation). Consideration of whether the reported face-part tuning would hold (or even be enhanced) if face-parts were presented in their typical locations should be included.

Our early visual cortex and some of the object-selective visual areas are sensitive to visual field locations. To dissociate the visual field tuning and face part tuning in face processing regions, in the main experiment of the current study the face part stimuli were presented at fixation to avoid the potential confounding contribution from visual field location. The spatial correlation between face part tuning and visual field tuning has been observed in posterior part of the face network (see response to point 4 above). It is unlikely that presenting the face parts at the fixation was responsible for the observed face part tuning. To directly test the role of stimulus location, we reanalyzed the data from control experiment 2 in which face parts were presented at their typical locations. Contrasting eyes above fixation vs. nose and mouth below fixation revealed similar anterior-posterior bias in the right pFFA, showing that the face part tuning in the right pFFA is invariant to the visual field location of stimuli. See comparison in Author response image 1, note that the maps of eyes on top vs. nose and mouth on bottom are unsmoothed.

**Author response image 1. sa2fig1:** 

6) Although several experiments (including two controls) have been conducted, each one runs the risk of being underpowered (n ranges 3-10). One way to add reassurance when sample sizes are small is to include analyses of the reliability and replicability of the data within subjects through a split-half, or other cross-validation procedure. The main experiment here consisted of eight functional runs, which is more than sufficient for these types of analyses to be performed.

Following the reviewer’s suggestion, we split the eight runs data from each participant in the main experiment into two data sets (odd-runs and even-runs), and estimated the eyes-mouth biases within each data set. Then we calculated the correlation coefficient between such biases across different voxels between the two data sets to estimate the reliability of the results in the right pFFA. The results demonstrate strong reliability of the data within participants. We have added these results in the Results session (Page 7 and Figure 2—figure supplement 1).

7) The current findings were only present within the right pFFA and right OFA. Although right lateralisation of face-processing is mentioned in the discussion, this is only cursory. A more expansive discussion of what such a face-part tuning might mean for our understanding of face-processing is warranted, particularly given that the recent work by de Haas and colleagues was bilateral.

The right lateralization of face-processing has been observed in face-selective network. Both the neural selectivity to faces (Kanwisher et al., 1997) and the decodable neural information of faces (Zhang et al., 2015) are higher in the right than in the left hemisphere. The neural clustering of face part tuning and consistent spatial patterns across individuals in the right rather than in the left face selective regions provides a potential computational advantage for right lateralization for face processing. The clustering of neurons with similar feature tuning have been found extensively in the ventral pathway, which may help to support a more efficient neural processing. Therefore, one of the neural mechanisms underlying the functional lateralization of face processing could be the existence of spatial clustering of face part tunings in the right hemisphere. We have added more discussion about the relevance between our results and lateralization of face processing (Page 25).

Reviewer #1 (Recommendations for the authors):While I think in general this is a well-executed study with powerful high-field 7T fMRI, I still have a few suggestions for the authors to consider.1. Clarify interpretation/motivation of the results: other than "faciotopy", why would neurons that are tuning to different face parts be spatially clustered? Given that BOLD signals are indirect measurements of neural activity, can the results about fmri voxels really suggest neuronal clustering? Would alternative interpretations such as activity synchronization be possible?

Replied in to Essential Revisions point 2.

2. The inclusion of a non-face object condition was nice. However, it might be helpful to clarify the logic behind regressing out response patterns from whole faces or everyday objects. Wouldn't it make more sense to regress out response patterns from the comparison between whole faces and everyday objects, i.e. faces – objects?

Replied in Essential Revisions point 3.

3. Testing functional connectivity profiles is a very interesting idea. Why not to take a look with the coverage of present data or perhaps datasets from openly available sources?

We thought about testing functional connectivity profiles of each cluster within each face-selective region, but unfortunately the FOV of our current data is too limited (only covers part of the occipito-temporal cortex) and varied quite bit across subjects, thus making it difficult to perform such a connectivity analysis. It is also very difficult to find openly available sources that (1) have high spatial resolution, (2) cover the whole brain or at least a large part of the brain, and (3) have well defined face ROIs especially with face parts preference identified. In our future studies, we’ll keep this in mind to try to obtain data that could support such a connectivity analysis.

Reviewer #3 (Recommendations for the authors):1) Can the authors make it clear whether the analyses concerning the P-A axis were conducted across voxels (i.e. volume space) or across surface vertices (i.e. surface space).

We are sorry for the unclear description. The P-A axis was defined across surface vertices. We have clarified this information in the manuscript (Page 8).

2) An uncorrected threshold pf p<0.01 for a functional localiser is non-standard and very liberal. Even with two-runs some form of reliability analysis can be performed.

The reason for choosing a liberal threshold of p<0.01 for ROI identification is to allow voxels potentially with different sensitivities to different face parts to be included in the current analysis. As we show in the results, the neural responses to nose, hair, and chin were lower than the responses to eyes and mouth. A strict threshold would likely exclude some of the voxels with lower sensitivity to certain face parts. Thus we used a relatively liberal threshold to include them in the analyses. We would like to note that with this threshold, the identified face-selective ROIs were highly responsive to whole-faces (Figure 1B, gray bars), and very selective to faces (Figure 3C).

3) Similarly, the main experiment comprises 8 functional runs. This amount of data can easily be split to perform reliability and reproducibility analyses. These would bolster the main claims as each experiment is relatively underpowered.

The reliability analysis results are shown in the Figure 2—figure supplement 1, see also our response to Essential Revisions point 6.

4) A pRF threshold of R2 = 0.02 (2%) is again very liberal. Can this selection be justified? Many prior pRF works even within FFA/PPA use a much higher threshold (~10-20%).

Generally, the R2 of pRF fitting is lower for FFA than OFA, as OFA is spatially closer to the early visual cortex. In our study, with R2 threshold of 10%, the number of identified voxels is much less for pFFA than for OFA, which makes it hard to observe any pRF spatial pattern in pFFA. Thus we chose a lower threshold for pRF analysis. Considering the aim of pRF analysis in the current study is to examine the contribution from visual field sensitivity to the observed face part pattern, rather than to examine the existence of the visual field sensitivity in pFFA, we believe it would not change our conclusion that the spatial pattern of face part tuning in pFFA could not be explained by the visual field tuning in pFFA.

References:

Cukur, T., Huth, A. G., Nishimoto, S., and Gallant, J. L. (2013). Functional Subdomains within Human FFA. *Journal of Neuroscience*, *33*(42), 16748–16766. https://doi.org/10.1523/JNEUROSCI.1259-13.2013

Kanwisher, N., McDermott, J., and Chun, M. M. (1997). The fusiform face area: A module in human extrastriate cortex specialized for face perception. *J Neurosci*, *17*(11), 4302–4311.

Kim, N. Y., Lee, S. M., Erlendsdottir, M. C., and McCarthy, G. (2014). Discriminable spatial patterns of activation for faces and bodies in the fusiform gyrus. *Frontiers in Human Neuroscience*, *8*. https://doi.org/10.3389/fnhum.2014.00632

Laughlin, S. B., and Sejnowski, T. J. (2003). Communication in Neuronal Networks. *Science*, *301*(5641), 1870–1874. https://doi.org/10.1126/science.1089662

Park, S. H., Russ, B. E., McMahon, D. B. T., Koyano, K. W., Berman, R. A., and Leopold, D. A. (2017). Functional Subpopulations of Neurons in a Macaque Face Patch Revealed by Single-Unit fMRI Mapping. *Neuron*, *95*(4), 971-981.e5. https://doi.org/10.1016/j.neuron.2017.07.014

Tsao, D. Y., Freiwald, W. A., Tootell, R. B. H., and Livingstone, M. S. (2006). A Cortical Region Consisting Entirely of Face-Selective Cells. *Science*, *311*(5761), 670–674. https://doi.org/10.1126/science.1119983

Weiner, K. S., and Grill-Spector, K. (2010). Sparsely-distributed organization of face and limb activations in human ventral temporal cortex. *NeuroImage*, *52*(4), 1559–1573. https://doi.org/10.1016/j.neuroimage.2010.04.262

Zhang, J., Liu, J., and Xu, Y. (2015). Neural Decoding Reveals Impaired Face Configural Processing in the Right Fusiform Face Area of Individuals with Developmental Prosopagnosia. *Journal of Neuroscience*, *35*(4), 1539–1548. https://doi.org/10.1523/JNEUROSCI.2646-14.2015